# Mutualist and pathogen traits interact to affect plant community structure in a spatially explicit model

John W. Schroeder[1,2 ✉], Andrew Dobson[3,4], Scott A. Mangan[1,5], Daniel F. Petticord[1] & Edward Allen Herre[1]

Empirical studies show that plant-soil feedbacks (PSF) can generate negative density dependent (NDD) recruitment capable of maintaining plant community diversity at landscape scales. However, the observation that common plants often exhibit relatively weaker NDD than rare plants at local scales is difficult to reconcile with the maintenance of overall plant diversity. We develop a spatially explicit simulation model that tracks the community dynamics of microbial mutualists, pathogens, and their plant hosts. We find that net PSF effects vary as a function of both host abundance and key microbial traits (e.g., host affinity) in ways that are compatible with both common plants exhibiting relatively weaker local NDD, while promoting overall species diversity. The model generates a series of testable predictions linking key microbial traits and the relative abundance of host species, to the strength and scale of PSF and overall plant community diversity.

[1] Smithsonian Tropical Research Institute, Balboa Ancon, Republic of Panama. [2] Department of Ecology, Evolution, and Marine Biology, University of California, Santa Barbara, Santa Barbara, CA, USA. [3] Princeton University, Princeton, NJ, USA. [4] Santa Fe Institute, Hyde Park Road, Santa Fe, NM, USA. [5] Washington University, St. Louis, MO, USA. ✉email: john.will.schroeder@gmail.com

Plant-soil feedbacks (PSF) are the processes by which plants influence the composition of the local soil microbiome (e.g. within the plant's rooting, or leaf litter zone), which, in turn, differentially affects the success of conspecific vs. heterospecific plant recruitment. Plant-soil feedback echoes the diversity maintenance mechanisms proposed by Janzen[1] and Connell[2], but shifts the focus from the effects of herbivorous and seed-eating insects to those of soil borne pathogens and mutualists. It is increasingly recognized that the strength and direction of PSF can profoundly affect plant community structure and diversity[3,4]. Negative PSF between two plant species occurs when the effect of a local build-up of pathogens reduces the recruitment of conspecific seedlings relative to heterospecific seedlings (see Supplementary Fig. 1, and Eq. (6), for an overview of the interaction coefficient, $I_s$, that measures plant-soil feedback). Theoretical studies have shown that negative PSF can promote the coexistence of competing plant species at large spatial scales by creating negative frequency dependent recruitment[5–8]. In contrast, positive PSF can result from a local build-up of beneficial microbial mutualists (e.g. mycorrhizal fungi or nitrogen-fixing rhizobia) that show moderate host affinity[9–11]. Positive feedbacks are expected to reduce species diversity[12]. In nature, net microbial PSF comprises the combined effects of pathogens (−) and mutualists (+)[13–15].

A key challenge to understanding how soil microbiomes affect plant community structure is correctly identifying the causes and consequences of interspecific variation in the strength of PSF[5,6,16,17]. Specifically, why do some plant species exhibit stronger negative PSF than others, and how does this variation relate to plant community dynamics? Following early studies, it has been proposed that common species should accumulate more pathogens and thereby suffer greater localized pressure from natural enemies[1,2]. However, multiple experimental studies have shown the opposite is apparently true; common plant species generally exhibit weaker negative PSF than rare species[6,13,18] (Fig. 1). Such studies have therefore suggested that the strength of PSF determines plant relative abundance; common species are common because they experience weaker net PSF effects. Given this apparent contradiction, the mechanisms underlying PSF effects on host species diversity and relative abundance require an explicit theoretical framework that will generate predictions for clear empirical tests.

Previous theoretical models of PSF have largely treated the host-associated soil microbiome of any given plant species and its resulting net effect on that host as a fixed, plant-associated trait[8,19]. This simplifying assumption ignores the empirical reality that plant microbiomes are not static, they are dynamic, changing with plant location, size and age[20–22]. It is therefore clear that the identity of the host plant indirectly affects the local microbial community composition by affecting the dynamics of the existing microbial community. Accordingly, factors such as the spatial arrangement of neighbouring plants, and their local species abundances, may influence the dynamics of the soil microbiota that mediate PSF[6,13,18]. The assumption that target host plant identity directly determines the local soil microbiome thus obscures potentially important mechanisms underlying the relationship between PSF and plant community structure. Crucially, if it is possible for the strength of PSF to affect host abundance, is it also possible that host abundance affects the strength of PSF by influencing the dynamics of soil biota?

Understanding what drives interspecific variation in feedback strength requires understanding microbial characteristics that affect population dynamics of the mutualists and pathogens that underlie feedbacks. These traits, such as host affinity and dispersal, may differ between microbes with mutualistic and those with pathogenic effects on their hosts. For example, studies comparing host association patterns of mutualists and pathogens suggest that putative pathogens show stronger affinity for particular hosts than mutualistic arbuscular mycorrhizal fungi[23–25]. In addition, both mutualists and pathogens can vary in dispersal propensity, from broad dispersal by wind[26,27], to limited dispersal via hyphal growth or belowground spores[26,28]. Thus, depending on which types of mutualists or pathogens predominate in a given system, the two guilds can differ in relative dispersal. This potential for pathogens and mutualists to differ in life history traits underscores the importance of considering how both mutualists and pathogens independently affect feedback.

In order to understand the implications of microbial community dynamics for plant community structure and diversity, we present a spatially explicit stochastic model that simulates trees interacting with their mutualistic and pathogenic soil symbionts through time. Departing from previous simulations, we use competition equations to control the change of the mutualist and pathogen communities through time on each host. We assume host plant identity does not directly determine microbial community composition; rather, plant identity affects the parameter values of the competition equations controlling microbial community dynamics. The resulting composition of mutualists and pathogens at a site determines seedling recruitment probabilities. In order to assess the importance of the role of PSF in maintaining plant diversity, we also assume that tree species can inherently differ in fitness, such that the fittest tree species would go to fixation in the absence of any stabilizing mechanisms generated by PSF. Using results of many simulation runs with random microbial trait values, we quantify the influence of each microbial trait on model outcomes, such as the strength and direction of PSF for each host species, as well as the overall diversity and relative abundance of host species (see Supplementary Note 1 for a brief explanation of our modelling philosophy).

Finally, we also use a combination of machine learning approaches (particle swarm optimization[29] and random forests[30]) to identify combinations of microbial trait values that (1) create a frequency dependent rare-species advantage that can maintain overall plant diversity, and (2) generate the commonly observed positive correlation between host abundance and PSF (i.e. common plants exhibit weaker negative local PSF). Using a general framework that directly links quantifiable microbial traits to observable outcomes, we develop testable mechanistic explanations for recent puzzling results, such as the existence of negative feedback driven by seemingly generalist pathogens[31,32], and the persistence of rare species even when they appear to be subject to stronger negative feedback than common species[5,6].

## Results

**Overview.** We determined the influence of different plant and microbial traits (illustrated in Table 1) on plant community dynamics by analysing the output of 16 K simulation runs with random, uniformly distributed values for our target traits. These model runs captured a wide range of simulation behaviours. We also used an adaptive learning algorithm[29] to identify parameterizations that (1) maintain diversity of microbes and trees through negative feedback, even in cases in which we assigned different fitness levels to different tree species, and (2) create a positive correlation between host abundance and PSF. After describing the dynamics of the simulation in the context of these specific requirements, we discuss the respective contributions of mutualists, pathogens, and dynamic (versus fixed) microbial communities to plant community dynamics. We then report the relative contributions of microbial traits to model outcomes.

We found that microbial community dynamics can create a correlation between host abundance and PSF, such that common

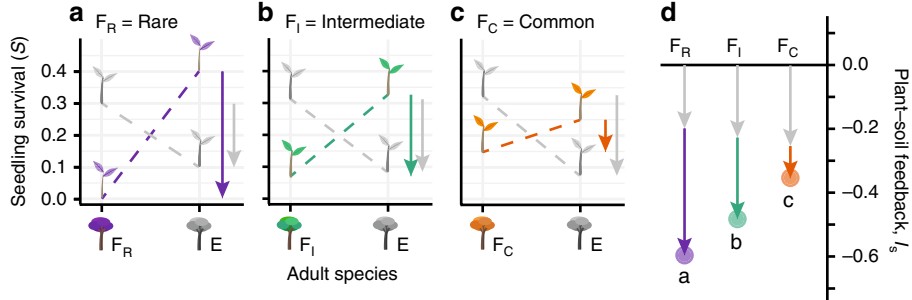

**Fig. 1 Plant-soil feedback of focal species (F) vs. all other species (E).** Plant-soil feedback can be calculated for a single species by taking the average of all pairwise interaction coefficients including that species[6,19]. Using this focal species interaction coefficient, and the home-away approach, studies have compared plant-soil feedback values between species that vary in abundance. In multiple systems, common species exhibit less negative PSF than rare species[6,13,56,57]. Above, we illustrate how measured values of seedling survival probabilities can create the relative decrease in the strength of negative PSF that common species often exhibit. Panels (**a**)–(**c**) show hypothetical values of seedling survival for rare, intermediate, and common species, that create a correlation between host relative abundance and the strength of PSF (panel **d**). This relative decrease in the strength of negative PSF for common species could either be caused by the accumulation of relatively less detrimental soil biota beneath common conspecifics (i.e. an increase in $S(a_A)$, Eq. (6)), or relatively more detrimental soil biota beneath heterospecifics of common plants (i.e. a decrease in $S(a_B)$, Eq. (6)). These two distinct possibilities are indistinguishable using metrics of PSF, or conspecific negative density dependence, but they would likely have different implications for plant community dynamics.

**Table 1 Explanation and summary of simulation parameters.**

| Parameter | Equation | Range | Fig. 2 value | Definition |
|---|---|---|---|---|
| $s_f$ | 1 | [0, 1] | $s_m = 0.35$<br>$s_p = 0.03$ | Host affinity of mutualists ($f = m$) and pathogens ($f = p$). |
| $\gamma_f$ | 2a | [0.1, 0.5] | $\gamma_m = 0.21$<br><br>$\gamma_p = 0.15$ | Fecundity of microbes, represented as a proportion of microbes in any given cell that are dispersed as propagules. |
| $b_f$ | 2b | [2, 3] | $b_t = 2.40$<br><br>$b_m = 2.82$<br>$b_p = 2.50$ | Exponent of power-law distribution. This indicates the dispersal limitation of both microbial guilds ($f = m$, $f = p$), or plants ($f = t$). |
| $g$ | 4 | [5, 15] | 8.80 | The impact of microbes on host survival. |
| $h$ | | [0.2, 1.5] | 0.40 | Relative contribution of the mutualist community to seedling recruitment probability. |
| $\zeta$ | | [0.8] | 0.8 | Relative fitness of the least fit plant species. The fitness values of plant species (i.e. $\zeta_j$ where $j$ is the plant species), are evenly distributed between $\zeta$ and 1. |
| $c_f$ | 5 | [0.5, 2] | $c_m = 0.78$<br>$c_p = 1.47$ | Exponent that scales the competitive ability of a microbe according to its effect on host survival. |
| $q_f$ | | [−0.5, 2] | $q_m = 0.91$<br>$q_p = 1.64$ | Exponent that scales intrinsic growth rate of microbes with host affinity. |
| $r_f$ | | [0.1, 2] | $r_m = 1.31$<br>$r_p = 1.64$ | Intrinsic growth rate of mutualists and pathogens. |
| $\alpha_f$ | | [0.5, 1.5] | $\alpha_m = 1.30$<br>$\alpha_p = 1.00$ | Competition coefficients of mutualists and pathogens. |

Parameter ranges represent the total range over which parameters were allowed to vary between runs. Values represent the values identified by a random forest model as most likely to drive a positive correlation between host abundance and feedback (i.e. those used in simulation runs depicted in Fig. 2). See Supplementary Table 1 for expanded parameter descriptions.

plant species experience weaker local negative PSF than rare plant species (Fig. 2). For example, when a new host colonizes a site, the pathogens with a relative affinity for that host rapidly increase in abundance (Fig. 3a), which decreases the relative survival probability of the focal host's seedlings (Fig. 3b). Concurrently, pathogens with affinity for other host species persist, slowly declining in abundance. When pathogens are not strictly host-specific (i.e. show some tendency to be generalists and persist on non-target hosts), it is possible for pathogens with a relative affinity for the most common host species to maintain higher abundances on non-target hosts than pathogens with a relative affinity for rare species (Fig. 2a). Therefore, seedlings of common species are expected to experience greater pathogen pressure beneath heterospecific adults (resulting in decreased survival probabilities) than seedlings of rare species (Fig. 2b).

In addition, mutualists with a relative affinity for common tree species as a primary host tend to reach relatively higher abundances beneath their hosts than mutualists with a relative affinity for rare trees do beneath their primary hosts. Thus, seedlings of common trees are expected to experience a greater benefit from their mutualists beneath conspecifics than seedlings of rare tree species (Fig. 2a). Acting together, the combined effects of the accumulation of pathogens of common tree species beneath heterospecifics, and the accumulation of their mutualists beneath conspecifics, each decrease the net strength of negative PSF experienced by common plants, resulting in a positive correlation between PSF and host abundance (Fig. 2c).

The dynamic mutualist and pathogen communities can also generate higher-order patterns of plant-soil feedback. Specifically, each plant species can have a different effect on the relative

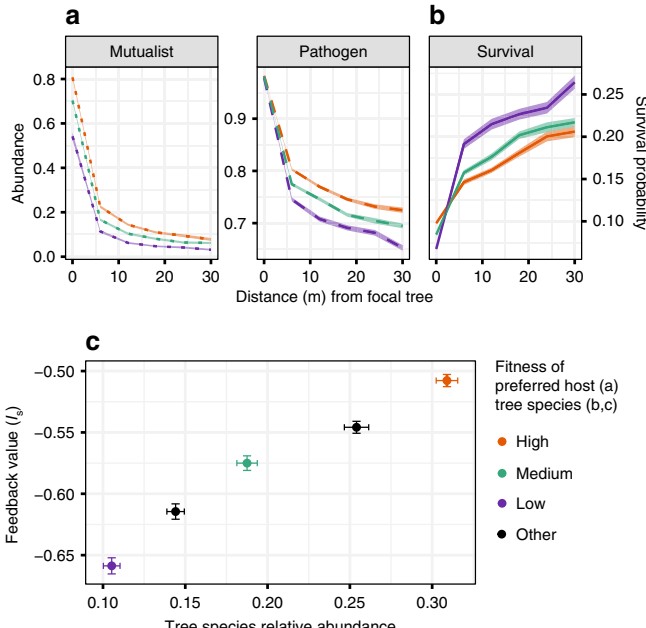

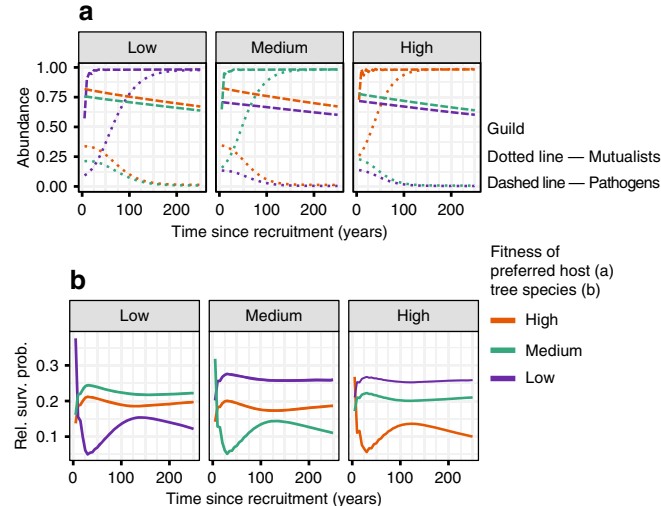

**Fig. 2 Spatial results from five-species simulation after 3 K time steps.**
Results are shown for three (of five) species representing the most fit, least fit, and intermediate species (high, low, and medium, respectively). Plant species achieve different abundances due to differences in fecundity (the low-fitness species produces 20% fewer seeds than the high-fitness species). **a** Distributions of pathogens and mutualists through space with respect to the distance from their preferred host. Abundances are relative to single-species carrying capacities. **b** Relative survival probabilities of seedlings of each of the three focal species through space with respect to distance from a conspecific. **c** Correlation between host relative abundance and feedback strength, $I_s$. Each point represents the mean final abundance and feedback strength of each species. Model parameters represent those that were most likely to show a strong positive correlation ($\rho > 0.8$) between host abundance and PSF according to the random forest classifier trained on data generated using particle swarm optimization. Shaded bands (**a**, **b**), and error bars (**c**) span mean values ± s.e.m., where $n = 48$ independent simulation runs. For (**a**) and (**b**), each independent value represents a mean value from a single simulation run. **a** shows that pathogens and mutualists are most abundant beneath their preferred hosts, and decrease in abundance with increasing distance from their host. Pathogens tend to maintain higher abundances on non-preferred hosts than mutualists. In addition, mutualists of common trees are more dominant on their preferred host than mutualists of rare hosts. Pathogens of common hosts maintain higher abundances on non-preferred hosts than pathogens of rare hosts, whence (**b**) common hosts are less likely to survive beneath heterospecifics than rare hosts. **c** As a result of the spatial distributions of survival probabilities driven by microbes, common trees experience less negative feedback than rare plants.

survival probability of the other species. For example, seedlings of the rare species are relatively more likely to survive beneath individuals of the intermediately abundant species than beneath individuals of a common species (Fig. 3b).

**Importance of mutualists and pathogens dynamics.** To determine the relative importance of mutualists, pathogens, and dynamic (versus fixed) microbial communities, we observed simulation dynamics after removing each of these features, respectively. When we remove mutualists or pathogens, or assume that microbial populations are fixed on each tree species, we observe qualitatively different plant population dynamics (Fig. 4a–e) that result in different frequency distributions of plant

**Fig. 3 Temporal results after 3 K time steps for five-species simulation.**
Results are illustrated for three species representing the most fit, least fit, and intermediate species (high, low, and medium fitness, respectively). **a** Mean abundances of pathogens and mutualists over time on each of the three host species. In all panels time = 0 represents the time at which the adult species occupying the focal cell changes to the species listed at the top of each panel. Each line represents the average population dynamics of the three mutualist and pathogen species as the focal plant increases in age. **b** Mean survival probabilities over time of each plant species in cells occupied by different species of adults. Each cut-out represents the average temporal dynamics of survival probabilities beneath one of the three focal adult species. Shaded bands in panels (**a**) and (**b**) cover the mean ± s.e.m., where $n = 48$ independent simulation runs. Although standard error values are displayed, values are too small to be visible. **a** Pathogens rapidly increase in abundance on their preferred hosts, but non-specific pathogens maintain high abundances as well. Conversely, mutualists increase more slowly than pathogens on their preferred host, and drive non-specific mutualists to low abundances. **b** After a new species recruits in a given cell, the survival probability of its seedlings rapidly decreases as its pathogen increases in abundance. Its survival probability subsequently increases with the abundance of its preferred mutualist.

species abundances (Fig. 4f–j). Figure 4a shows that feedbacks between trees and their soil mutualists and pathogens can allow tree species to coexist at different stable equilibria over long periods of time (despite intrinsic fitness differences), leading to a skewed normal distribution of abundances (Fig. 4f). When pathogens are removed from the simulation, negative PSF is lost and the system moves to a monoculture with fixation of the fittest tree species (Fig. 4b, c). In contrast, when mutualists are removed from the simulation, all host species coexist, and the variation in equilibrium abundance values among plant species decreases (Fig. 4d).

In addition to assessing the effects of the presence or absence of pathogens and mutualists, we also assessed the importance of spatial and temporal variation in microbial communities associated with each host. We did this by fixing microbial communities at the species-specific averages (as has been assumed in previous plant community models[5,6]) that were generated after the equivalent of 4 K years of simulation runs (i.e. stable plant communities). Once microbial populations are fixed, plant-soil feedback provides a much weaker stabilizing effect, and rare plant species frequently go locally extinct in the simulation (Fig. 4e). Thus, the spatial and temporal dynamics of microbial communities emerges as a potentially key factor in promoting host species diversity.

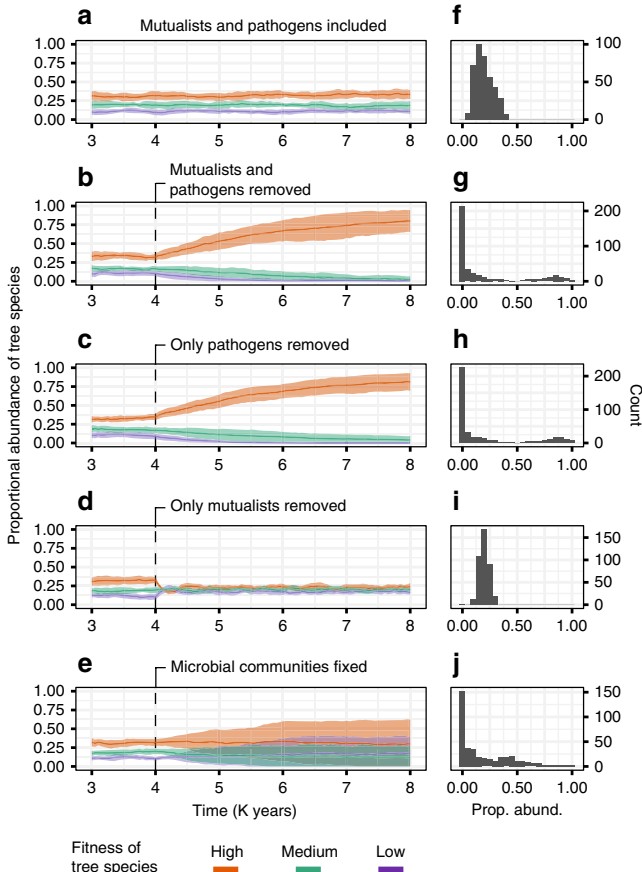

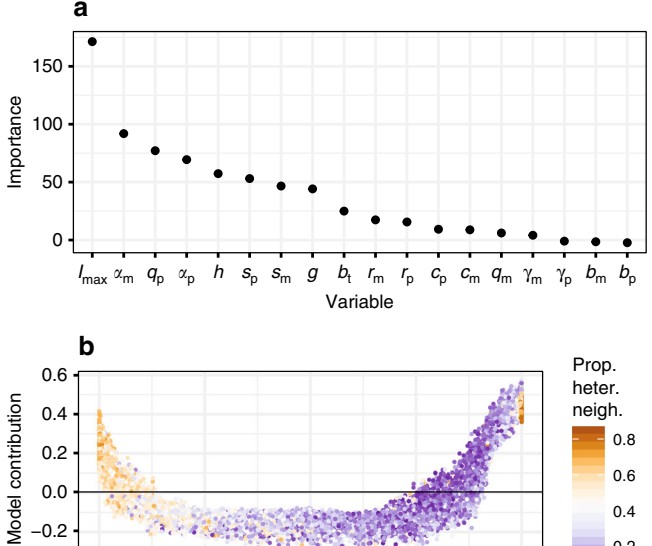

**Fig. 5 Predictors of plant coexistence according to random forest.** Results from a random forest classifier using each variable (including maximum net PSF value, $I_{max}$) to predict whether all five species coexist until the end of the simulation. **a** Importance of each variable in predicting plant diversity maintenance, where importance is the mean decrease in the model's classification accuracy when a variable is permuted (normalized by the standard deviation). **b** Conditional feature contribution of PSF strength to plant diversity. Each point represents the relative contribution of $I_{max}$ to the predicted value of plant diversity of an independent model run. A model contribution value >0 means that the feedback value for that run contributes positively to the prediction of species coexistence. Point colours indicate the proportion of heterospecific neighbours for each tree at the end of the run. A value of zero represents a forest dominated by one species (100% conspecific neighbours), and a value of 1 means that no two adjacent plants are of the same species. Panel (**b**) shows that negative feedback maintains diversity, and is associated with spatial dispersion of plants. Strong positive feedback can also maintain diversity by reducing species turnover, but this represents an extreme case unlikely to be stable through time. See Supplementary Fig. 2 for the conditional feature contributions of every variable.

**Fig. 4 Plant population dynamics with and without microbial dynamics.** Panels (**a**) through (**e**) each show the average population dynamics of plant species. Average dynamics (mean abundance ± s.d. encompassed by each shaded band, $n = 16$ independent simulation runs) are shown only for the three plant species with high fitness, medium fitness, and low fitness. Panels (**f**) through (**j**) show the resulting frequency distributions of plant abundance, which include all five species from 16 runs. For panels (**f**) through (**j**), abundance values were pooled across runs. **a** Runs were conducted using the parameterization shown in Table 1. **b** Runs conducted as in (**a**), but after 4 K years, all effects of mutualists and pathogens on plant recruitment were removed from the simulation. Panel (**b**) therefore represents the dynamics expected without plant-soil feedback; the species with highest fitness goes to fixation. **c** All pathogens were removed after 4 K years. Feedback dynamics without pathogens are similar to those with no feedback. **d** All mutualists were removed after 4 K years, but pathogens were kept in the system. Without mutualists, the most abundant and least abundant plant species shared more similar equilibrium abundances. **e** Microbial communities were fixed on each plant species. I.e. after 4 K years, each new recruit of a given plant species was assigned a specific microbial community representing the mean microbial community on individuals of that host species after 4 K years. These fixed microbial communities did not change through time once assigned. Panel (**e**) illustrates the dynamics that would be expected if each plant species supported a species-specific microbial community that does not change through time. Population dynamics in two dimensions are expected to occur at a faster rate than the one-dimensional population dynamics displayed here[51].

**Relationships between microbial traits and simulation dynamics.** Conducting many individual simulations (16 K) that were parameterized using random values of our target variables allowed us to characterize the importance of each model parameter (including microbial traits) as a determinant of model behaviour. Here, we report the results of three random forest classification models[30] that measure the contribution of

simulation parameters to (1) the maintenance of plant species diversity, (2) the development of strong negative PSF, and (3) a correlation between host abundance and PSF. The first random forest (accuracy = 0.88, precision = 0.87, recall = 0.71) identified plant-soil feedback strength (defined in Eq. (6)) as the primary determinant of plant diversity maintenance (Fig. 5, Supplementary Fig. 2). According to the random forest feature contribution plot, strong negative and positive feedback are both associated with the maintenance of diversity over the course of the simulation; strong negative feedback tends to generate more even spatial distributions of plants, while positive feedback is associated with strong spatial aggregation (Fig. 5b). Furthermore, plant abundances are much more likely to reach a stable temporal equilibrium under negative feedback than under positive feedback (Supplementary Fig. 3). Thus, the simulation indicates that positive feedback would not maintain diversity indefinitely.

A second random forest classification model identified simulation parameters that determine whether strong negative feedback develops (i.e. whether $I_{max} < -1.5$; accuracy = 0.94, precision = 0.79, recall = 0.92). The strongest predictor of

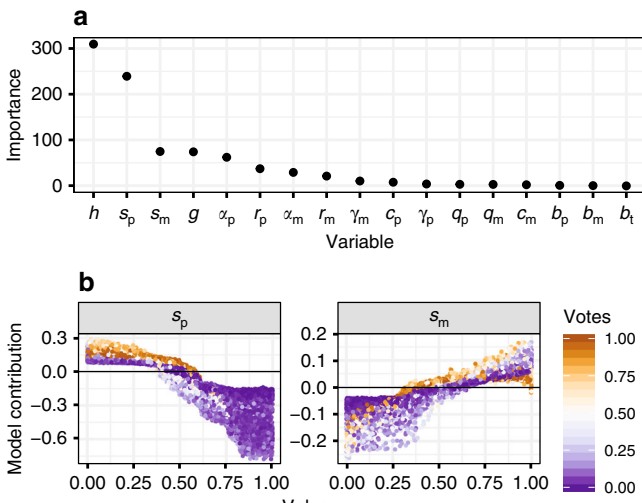

**Fig. 6 Predictors of strong negative feedback according to random forest.** Results from a random forest classifier using each variable to predict whether strong negative feedback develops. **a** Importance of each variable as a predictor of the development of strong negative PSF (true if $l_{max} < -1.5$, and false if $l_{max} > -1.5$). **b** Conditional feature contributions of mutualist and pathogen host affinity as predictors of whether strong negative feedback will develop. Point colour represents the proportion of decision trees that predict that strong negative feedback will develop. See Supplementary Fig. 3 for the conditional feature contributions of every variable.

negative feedback is the relative effect of mutualists on seedling survival, $h$ (Eq. (4)), followed by the relative host affinity of pathogens, $s_p$, and mutualists, $s_m$ (Eq. (1), Fig. 6a, Supplementary Fig. 4). As the host affinity of pathogens increases (i.e. as the proportional effect of pathogens on the survival of non-preferred hosts decreases), negative feedback is more likely to develop (Fig. 6b). Conversely, as the host affinity of mutualists increases, the probability that negative feedback will develop decreases. Negative feedback can still develop when pathogens show only moderate host affinity (Fig. 6b).

A third random forest classification model (accuracy = 0.87, precision = 0.04, recall = 0.72) determined that a key variable predicting whether a positive correlation between plant abundance and feedback will develop is $q_p$ (Eq. (5)), the parameter scaling a pathogen's intrinsic growth rate with its effect on a host (Supplementary Fig. 5, Supplementary Fig. 6). When $q_p$ increases, pathogen populations decline more slowly on non-preferred hosts. Thus, when $q_p$ is higher, pathogens persist at a site after their host dies (e.g. population dynamics illustrated in Figs. 2–4). In this way, pathogens of common plant species maintain higher abundances on non-preferred hosts than pathogens of rare plant species. Notably, $q_p$ is much more important as a predictor of a positive correlation between host abundance and PSF strength than pathogen dispersal propensity, $b_p$ (Eq. (2b)), or fecundity, $\gamma_p$ (Eq. (2a)). This shows that pathogen persistence, rather than more recent spill-over from nearby common hosts, primarily drives the correlation between host abundance and PSF strength.

## Discussion

Understanding the mechanisms underlying plant community diversity and structure is central to understanding drivers of terrestrial biodiversity[33]. Previous work has explored the role of plant-soil feedback (PSF) as a process structuring plant communities[6,13,19], but a lack of detailed mechanistic PSF models has limited our understanding of the causes and consequences of

PSF for plant community structure. The general mechanistic model described here explicitly simulates the community dynamics of soil symbionts and uses the resulting microbial community structure to determine plant recruitment probabilities. We use this simulation framework to disentangle the relationship between observed PSF and the microbial life history traits that create these effects: dispersal, persistence, host affinity, and competitive ability of microbial pathogens and mutualists. Specifically, we find (1) negative PSF maintains diversity and spatial dispersion of plant species, (2) microbial traits, particularly the relative host affinity and effect of mutualists and pathogens on plant survival, are key to determining whether negative PSF develops, and (3) mutualist and pathogen community dynamics can interact to produce the observed positive correlation between host abundance and PSF in a way that also contributes to stable coexistence of plant species. These insights only become apparent when we allow microbial communities to change dynamically. By explicitly linking microbial ecology with plant-soil feedback, we make testable predictions about how quantifiable microbial life history traits correspond to observable aspects of plant and microbial diversity and community structure.

Our findings corroborate previous experimental, demographic, and theoretical results by showing that negative feedback maintains overall diversity and promotes spatial dispersion of plants[6,8,19]. Building on this previous work, we determine how observable microbial traits and dynamics may produce such feedback patterns. For example, one salient result from our simulation was the potential importance of mutualist traits in affecting PSF. Specifically, we found that the host affinity of pathogens and mutualists both strongly influence the direction of PSF (more host-specific pathogens drive negative feedback, and more host-specific mutualists drive positive feedback). In tropical forests, where PSF has been observed to be negative[6], plant-associated mutualists appear to be less host-specific in their association patterns than pathogens[9,23,24]. This topic deserves more attention, ideally empirical quantification of the effects of mutualists and pathogens on host growth and survival, and how these effects differ between host plant species and environments. Theory provides some important clues as to how these traits are expected to vary between environments. For example, Thrall et al.[34] predict that pathogens more strongly affect host survival with increasing resource availability, while mutualists become increasingly host-specific (and pathogens less-so) with increasing host diversity. Unfortunately, direct comparisons between characteristics of mutualists and pathogens are rare or lacking. Everything described above emphasizes the need for these studies.

Our simulation framework makes predictions about the consequences of such variation in microbial traits for forest dynamics. Monodominance in some forests dominated by ecto-mycorrhizal tree species, such as the Ituri forest in the Congo, is one example[35,36], but see ref. [37]. Here, an increased role of mutualists in seedling survival is expected to create strong positive feedbacks that, in turn, promote monodominance. Environmental conditions may also affect the relative importance of mutualists vs. pathogens in determining plant recruitment more generally. For example, mycorrhizal associations facilitate abiotic stress tolerance in plants, such as drought and salt tolerance[38,39]. Where environmental stressors shift the primary drivers of recruitment limitation from pathogens to mutualists, our simulation predicts that feedbacks will be more positive, thereby decreasing local plant diversity, and increasing spatial aggregation of hosts. Conversely, where key mutualisms are potentially less important, such as in areas of high concentrations of particularly relevant resources[34], feedback strength should be more negative, as suggested by La Manna et al.[40], thereby increasing plant diversity and spatial dispersion of hosts. This and related

propositions can, and should, be explicitly tested in a variety of plant communities.

Recent work has proposed that variation in plant susceptibility to pathogens (e.g., innate genetic differences in pathogen resistance) drives the observed correlation between host abundance and plant-soil feedback strength[5,6,17]. Our work expands these initial insights, and provides a vital additional mechanism by which common plants may develop less negative PSF than rare plants. As plants become more common, their pathogens become relatively more abundant beneath heterospecifics than those of rare plants, thereby reducing the strength of net negative feedback for common plants, and reducing their survival probability. Concomitantly, common plants accumulate more of their preferred mutualists than rare plants, increasing survival probability beneath conspecifics, and decreasing the strength of negative feedback. Even when different plant species are equally susceptible to their symbionts (both pathogens and mutualists), microbial community dynamics drive a positive correlation between overall host abundance and local plant-soil feedback. Thus, while many factors can affect the overall abundance of plants, it is possible that a positive correlation between host abundance and PSF will develop largely due to the effects of microbial community dynamics.

Determining whether the mechanisms we describe above can explain interspecific differences in PSF strength has important implications for plant community dynamics. In the simulation presented here, as species become rarer, the survival probability of their seedlings beneath heterospecific trees increases relative to common species, effectively generating a rare species advantage. This strong negative feedback among rare plants provides a buffer against extinction[41]. Conversely, a decrease in survival probability beneath conspecifics would undermine this buffer against extinction for rare species[41]. So far, too few studies have explicitly addressed the dichotomy of whether interspecific variation in PSF is dominated by interspecific variation in survival beneath conspecifics versus survival beneath heterospecifics but see ref. [42].

Our results provide predictions that can be tested to determine whether feedback strength tracks host abundance. According to our simulation, if the relative abundances of certain plants are artificially increased for long periods of time, their pathogens will increase in abundance on non-target hosts. If this is the case, then plant species that have been selectively cultivated (e.g. tree species domesticated in pre-Columbian America[43,44]), should exhibit relatively weaker negative feedbacks in areas where they have been selectively cultivated. Measuring plant-soil feedback along gradients of selective cultivation provide tests of the spatial and temporal scales over which the microbial communities that affect PSF track host abundance.

The suite of microbial traits required to drive the correlation between feedback and host abundance in our simulation (e.g., strong functional host specificity, but an ability to persist on all hosts) also provides potential explanations for some apparently counterintuitive results. Our results show that strict host specificity is not necessary for PSF effects to maintain host diversity. Indeed, the degree to which pathogens persist on non-target hosts in our simulation could be interpreted as an indication of generality. Many pathogens appear to be host generalists in nature[45,46], despite the fact that they often drive negative plant-soil feedbacks[47]. This result is relevant to the interpretation of empirical studies that increasingly show that functionally specific pathogens with broad host ranges are the norm[32,48]. An additional possibility is that such generalist pathogens rapidly adapt or evolve to exploit a host occupying a given site[49]. In this case, the model results referring to microbial abundance could be interpreted as the degree of adaptation to a specific host species. Future studies should determine the degree to which microbial

plant-soil feedbacks are driven by shifts in the abundances of different microbial species, and/or rapid adaptation of these same microbial species to particular hosts.

In conclusion, here, we develop and present a general model to investigate the mechanistic relationship between microbial community dynamics and plant community structure and diversity. The simulation corroborates previous studies in identifying negative plant-soil feedback (PSF) as a driver of plant diversity maintenance. Using a machine learning approach, we also identify key aspects of the simulation (including microbial traits) that most affect the dynamics of PSF, such as the relative effect and host affinity of pathogens and mutualists. Furthermore, by relaxing the assumption that soil microbial communities are fixed by host-plant species identity, we find that microbial community dynamics can explain puzzling and paradoxical patterns, such as a correlation between host abundance and PSF. The model presented here offers a flexible mechanistic framework that significantly expands the classic Janzen-Connell model of plant diversity by directly connecting microbial ecology to plant community dynamics. It will be intriguing to determine if similar mechanisms can explain the maintenance of biodiversity in other hyper-diverse systems such as coral reefs and bacterial communities.

## Methods

**Simulation overview**. All simulations take place in a one-dimensional circular array, occupied by a specified number of trees ($n$), each with a unique community of root-associated microbial pathogens and mutualists. We chose to conduct the simulations in one dimension to simplify interpretation, but we recognize that simulation dynamics likely progress more slowly than they would in two dimensions[50]. At each time step, the simulation randomly selects a subset of the tree population for replacement. All trees produce a specified number of seeds at each time step that disperse according to a power-law dispersal kernel. The probability that an individual of a given tree species recruits to the adult population at a site is a function of the community of microbial propagules (pathogens and mutualists), and the relative abundance of the tree species' seeds, occurring at the site. A microbial community is then calculated for the recruit based on the abundances of the microbes at its location, the compatibility between the microbes and the species of recruit, and the competitive ability of the microbes. At each time step, the populations of microbes change on each adult according to competition equations defined below. The simulation continues for an arbitrary number of time steps.

**Assumptions**.

1. All plant species are equivalent, except in the identity of their microbial pathogens and mutualists, and the absolute survival probability of their seedlings (a representation of fitness differences that is unrelated to interactions with microbes). In the absence of plant-microbial interactions, plant communities develop according to neutral expectations.
2. Plants, mutualists, and pathogens are all dispersal limited, and follow a power-law dispersal function. We allow the scale parameter (indicating dispersal distance) of the dispersal kernel to vary between plants, pathogens, and mutualists, respectively.
3. Microbial taxa (both pathogens and mutualists) have two host compatibility levels: one value for their primary host, and another value—less than or equal to the value for the primary host—for all other host species. This means that all non-target hosts are equivalent for each microbial taxon. Thus, in the simulation, pathogens and mutualists can be complete host specialists, complete generalists, or capable of associating with all hosts to varying degrees. Empirical results suggest that many pathogens and mutualists may associate with a broad range of hosts, but that they have host-specific effects[32].
4. Mutualists are beneficial, and pathogens are detrimental to host survival.
5. Each microbial taxon has an identical carrying capacity on each host.
6. The effect of a microbial taxon on its host is correlated with the competitive ability of the microbial taxon on that host (i.e. more beneficial mutualists outcompete less beneficial mutualists, and more detrimental pathogens outcompete less detrimental pathogens)[51–53].
7. Once a tree becomes an adult, its microbial community is subject to change according to internal dynamics, and immigration of microbial propagules from nearby trees. We choose to model these changes according to competition equations with logistic growth.

**Simulation description**. Each simulation starts as a circular array populated with $n$ trees, of $k$ species, evenly positioned through space. Each tree species is the

preferred host for one mutualist species and one pathogen species, but concomitantly has the potential to associate with all pathogens and mutualists to varying degrees defined by the simulation. Thus, the mutualist and pathogen community associating with each tree is described by a vector of abundances.

For each microbial taxon, the degree of affinity, $s$, ranges from zero (complete specialist that only generates an effect on preferred hosts) to one (complete generalist that generates an equal effect on all hosts). All microbial species in a given guild, $f$ (mutualists, m, and pathogens, p), have a guild-specific specificity value, $s_f$, such that the compatibility of a microbe $i$ with host species $j$ is:

$$e_{fij} = \begin{cases} 1, & j = \text{preferred host} \\ s_f, & j \neq \text{preferred host} \end{cases} \qquad (1)$$

The probability that the seedling survives to recruit successfully to the adult population is determined by the microbial community at the location to which it disperses, and the compatibility between the microbial taxon and the candidate seedling. The microbial community at any given location, $F_x$ (mutualists, $M_x$, or pathogens, $P_x$) is the sum of the microbial abundances occupying the adult in the cell at time $t$, and the microbes arriving as propagules. The density of pathogen and mutualist propagules at a given location are determined by taking the sum of the microbial communities of all adults, scaled by their distance from the location according to a truncated discrete power-law distribution:

$$F_x(t+1) = F_x(t) + \gamma_f \sum_{j=1:n} F_{h_j}(t) \cdot \text{powerlaw}\left(\left|x - \text{pos}\left(h_j\right)\right|\right) \qquad (2a)$$

$$\text{powerlaw}(x) = \frac{(1+x)^{-b_f}}{\sum_{n=0}^{\infty}(1+n)^{-b_f}} \qquad (2b)$$

where $\text{pos}(h_j)$ corresponds to the position of host $j$, and $\gamma_f$ represents the fecundity of the mutualists or pathogens. The mutualist community, $M_x$, and the pathogen community, $P_x$, are used to calculate the recruitment probability of the seedling. At each recruitment trial, this compatibility score is used to calculate the impact of a given microbial taxon on the probability of recruitment of its host. For mutualists and pathogens, the impact is equal to the sum of the abundance of each microbial taxon multiplied by its compatibility with the focal host. So, for a given seedling of species $j$, at location $x$, the effect of the microbial community on the seedling is:

$$\theta_f(x, j) = F_x \cdot e_{fj} \qquad (3)$$

where $F_x$ represents the vector of microbial abundances at location $x$, and $e_{fj}$ represents the vector of effect values of microbes on species $j$. The total effects of pathogens and mutualists determine the recruitment probability, $\Pi$, of a seedling via the following function:

$$\Pi\left(\theta_m, \theta_p, j\right) = \frac{\zeta_j \text{logistic}\left(g\left(h\theta_m - \theta_p\right)\right)}{\sum_{n=1:k} \Pi\left(\theta_m, \theta_p, n\right)} \qquad (4)$$
$$\text{logistic}(x) = \frac{1}{1 - e^{-x}}$$

where $\zeta_j$ represents the relative fitness of plant species $j$. The coefficient derived from Eq. (4) is calculated for each plant species arriving at an empty cell, and standardized by the number of seeds of each species landing in the cell. The successful recruit is then drawn from the resulting multinomial distribution.

The recruit becomes a reproductive adult in the next time step, and begins modifying the abundance of each microbial taxon through time according to the following differential equation:

$$\frac{dF_{ij}}{dt} = r_f e_{fj}^{q_f} F_{ij}\left(1 - \left(F_{ij} + \sum_{n \neq j} \alpha_f e_{fn}^{c_f} F_{in}\right)\right) \qquad (5)$$

For each microbial guild, $r_f$ represents the intrinsic growth rate when associating with its preferred host. When associating with a non-preferred host, we multiply by the effect of the microbe on non-target hosts, $s_f$, raised to the exponent $q_f$. The parameter $\alpha_f$ represents the competitive ability of a microbial taxon when associating with its preferred host. The competition coefficient is multiplied by $s_f^{c_f}$ when the microbe is associated with a non-preferred host. The microbial community changes on each tree in the manner described by Eq. (5) at each time step. This cycle of mortality and replacement continues for a predetermined number of time steps.

**Measuring plant-soil feedback.** To offer results that are directly comparable to empirical studies, we tracked plant-soil feedback, the process that drives negative density dependence. We measured plant-soil feedback strength using Bever's interaction coefficient, $I_s$ (See Supplementary Fig. 1)[19,47]. The metric can be calculated for any two plant species, A and B, as follows:

$$I_s = S(a_A) + S(b_B) - (S(a_B) + S(b_A)) \qquad (6)$$

where $S(a_A)$ and $S(b_B)$ are the respective mean survival probabilities of seedlings of species A and B grown beneath conspecific adults (home performance), and $S(a_B)$ and $S(b_A)$ are the mean survival probabilities of seedlings of species A and B grown beneath adults of the other species (away performance). A given plant species' value for $I_s$ is the mean of all pairwise feedback values that include the focal species.

When using $I_s$ as a predictor of model outcomes (e.g. for random forest models), we measured $I_s$ at the beginning of the simulation. Specifically, we measured $I_s$ after allowing microbial communities to change on their hosts following Eq. (5) for 50 time steps without host mortality. To quantify the relationship between host abundance and PSF (at the end of each simulation run), we calculated the Spearman correlation coefficient between PSF and host abundance.

**Determining whether simulations reached equilibrium.** To establish whether plant abundances reached a stable equilibrium at the end of the simulation, we employed an equilibrium metric, $P_e$. To calculate $P_e$, we measured the change in abundance of the most common plant species across all windows of 100 time steps in the 600 time steps leading up to the end of the simulation (subsampled to include every 10th time step in order to improve computational speed). We then used a $t$-test to determine whether the average rate of change across 100 time steps deviated from zero. The equilibrium metric, $P_e$, is the $P$-value of this two-sided $t$-test. Higher values of $P_e$ indicate that the simulation is more likely to have been at a stable equilibrium over the final 600 time steps.

**Characterizing simulation behaviour.** Given the large number of possible parameter value combinations, we used a random search to explore parameter space, then characterized model behaviour using a random forest classification approach[30]. For each run, parameter values were drawn from independent random uniform distributions defined in Table 1.

Random forest classification was used to construct models predicting whether (1) all plant species coexisted throughout the simulation, (2) whether strong negative feedback developed (i.e. the maximum strength feedback was less than −1.5), and (3) whether a strong positive correlation between host abundance and plant-soil feedback developed ($\rho(I_s) > 0.8$, $\sigma(I_s) > 0.02$, $I_{max} < 0$, and richness of plants, mutualists, and pathogens = 5). For each model, all parameter values were included as predictors. For model (1), we also included maximum PSF strength, $I_{max}$, as a predictor. Sample sizes for random forest fitting were balanced, such that both classes were equally represented. Hyperparameters for each random forest fit were otherwise set to the default settings for classification using the R package randomForest[54].

For each random forest model, we estimated the importance of each parameter as a predictor of model output. We quantified variable importance as the decrease in prediction accuracy resulting from sample-wise permutation of the values among out-of-bag samples. I.e. if removal of a parameter from the classification model decreased the model's prediction accuracy, the parameter was deemed important. Feature contributions of each model parameter were visualized as the out-of-bag cross-validated conditional contributions of each parameter value to the predicted model outcome[55]. In feature contribution plots, each point represents the average (among decision trees) contribution ($y$) of a parameter value ($x$) to the predicted outcome. Random forest analyses were conducted using the R package randomForest[54], and the variable importance and contributions were estimated using the R package forestFloor[55].

**Simulation runs.** We ran 16,000 simulations, each with five species of plants comprising 499 individuals. Parameter values were drawn randomly from the ranges listed in Table 1. For each simulation run, the starting location of each individual tree was randomized, and each tree and microbial species started at equal total abundances. At each time step, ten percent of the adult trees were replaced (equivalent to a 5-year time step, assuming two percent mortality yr$^{-1}$). Simulations were run for 3 K total steps (equivalent to 15 K years). See Figs. 2 and 3 for an overview of simulation output.

**Optimizing for correlation between abundance and feedback.** The probability of randomly achieving a simulation parameterization that matched any set of PSF patterns was low, because we explored a very broad range of parameter values. This broad search allowed us to clearly understand the independent influence of each parameter on simulation results using random forest models, but also meant that any specific simulation result was poorly replicated in our random search. We therefore used particle swarm optimization (PSO) to identify combinations of parameter values most likely to simultaneously promote species coexistence under negative PSF, and generate a positive correlation between PSF and plant abundance (i.e. $\rho(I_s) > 0.8$, $\sigma(I_s) > 0.02$, $I_{max} < 0$, and richness of plants, mutualists, and pathogens = 5).

In PSO, combinations of parameter values are described as particles, and the population of particles is called a swarm. To update particle positions at each iteration, $t$, we first employed the following equation to determine particle velocity, $v$, of each particle $x_i$ in each dimension $d$:

$$v_{id}(t+1) = w \cdot v_{i,d}(t) + c \cdot \text{rand} \cdot (p_{id} - x_{id}) \qquad (7)$$

where $w$ and $c$ are constants (we selected 0.5, and 0.8, respectively), rand represents a random number drawn from a uniform distribution bounded by 0 and 1, and $p$ represents the position of the best solution. At each step, we trained a new random forest classification model, then took the 10 solutions deemed most likely to produce negative feedback and the feedback-abundance correlation. For each particle in the swarm, we randomly selected one of the 10 best solutions to

determine $p_{id}$, then updated the position of each particle was determined with the following equation:

$$x_{id}(t+1) = x_{id}(t) + v_{id}(t+1) \tag{8}$$

At each time step, we randomly selected four particles to replace with random parameterizations to limit premature convergence. To ensure that we were not identifying a local optimum, we iteratively conducted PSF, starting with 16 non-overlapping optimizations. For each optimization, we allowed a swarm of 50 particles to iteratively explore the parameter space for 20 PSO steps, starting from random positions. We then combined all results from two optimization runs (i.e. eight combinations of 1000 total runs), and conducted eight PSO optimizations for another 20 time-steps. We continued this process until the final optimization included all simulation runs. This optimization process allowed us to confirm that independent optimizations identified similar solutions, rather than different local optima. We then used a final random forest classification model to identify the parameterization most likely to produce negative feedback and the feedback-abundance correlation among all runs conducted throughout the PSO optimizations (Figs. 2–4).

**Isolating effects of model components on population dynamics**. To characterize the importance of mutualists and pathogens in determining plant community dynamics, we conducted additional runs in which we removed the effect of (1) all microbes, (2) only pathogens, and (3) only mutualists, while tracking plant abundances through time. Specifically, after letting the simulation run for an equivalent of 4 K years, we removed each of the following model components from independent batches of 16 runs: (1) the effect of microbes on seedling survival (i.e. to simulate neutral dynamics); (2) only the effect of pathogens; (3) only the effect of mutualists.

For an additional batch of 16 runs, we fixed the microbial community composition on each plant species. At the simulation time step equivalent to 4 K years, we calculated the mean abundances of each microbial taxon on individuals of each host plant species. These species-specific average microbial communities were subsequently assigned to all new recruits for the rest of the simulation. This latter approach is analogous to the approach used in simulations that assume trees have a fixed effect on the survival of conspecific vs. heterospecific seedlings[5,8,19].

**Reporting summary**. Further information on research design is available in the Nature Research Reporting Summary linked to this article.

## Data availability
Simulation results that support the findings of this study are archived under https://doi.org/10.5281/zenodo.3742143, and can be accessed on GitHub (see below).

## Code availability
All R code required to run the simulation described here is available on GitHub at https://github.com/johnwschroeder/PlantMicrobeSimulation, and archived under https://doi.org/10.5281/zenodo.3742143. The code is available for reuse under an MIT License.

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

## Acknowledgements

We gratefully acknowledge support from the Smithsonian Tropical Research Institute through a Smithsonian Institution Postdoctoral Fellowship (J.W.S.) and a STRI Short-term Fellowship (D.F.P.). We also thank the Hillary Young lab at UCSB, as well as Kirk Broders, Holly Moeller and Helene Muller-Landau, for valuable feedback during the preparation of this paper.

## Author contributions

J.W.S. wrote the simulation, conducted the analyses, and drafted the paper with input from all coauthors. A.P.D., S.A.M., D.F.P. and E.A.H. contributed substantial revisions to each draft of the paper.

## Competing interests

The authors declare no competing interests.
