## [Peer Review File · Nature Communications]

Reviewers' Comments:

Reviewer #1:

Remarks to the Author:

In this manuscript, the authors used simulation models to explore how changes in the traits of microbial pathogens and mutualists influence plant-soil feedback. Specifically, they were interested in determining what scenarios could resolve an apparent "paradox" in the world of plant-soil feedback – how to reconcile negative plant-soil feedback as a coexistence mechanism with the observation that common plant species often have weaker negative feedbacks. While most simulations of plant-soil feedback treat feedback as a fixed trait of the plant, these simulations allowed the composition of mutualist and pathogen communities to change based on composition of the surrounding microbial communities, competition among microbes, and affinity for host plants. The authors found support for negative plant-soil feedback generating coexistence among species and found that allowing different pathogen and mutualist dynamics can generate negative plant-soil feedback and the positive correlation between feedback and abundance.

Given the interest in plant-soil feedback as a coexistence mechanism and the conundrum of negative feedback generating coexistence but common species having weaker negative feedback, I suspect this paper will be of interest to many plant community ecologists. While I am not an expert on the methods, they seem sound and generated interesting, plausible results. The discussion is full of exiting ideas that can be empirically tested. My main critique is that I believe more details on plant-soil feedback and the model should be added to the introduction to let the reader follow along without having to dig into the methods.

(L75) I know there are space limitations, but it would help place the work in context to provide a few key empirical examples of how mutualistic and pathogenic microbes differ in traits.

(L83) "allow mutualist and pathogens communities to dynamically change beneath each tree following competition equations" – From the introduction, it is still unclear how the simulations are allowing microbial communities to change. Earlier (L63-66) made it seem like it was going to be a function of the surrounding plant community. Here, it seems like it is a function of microbial competition. After reading the methods, it is clear that it is a function of both factors, along with the compatibility between the microbes and the host. Clearly stating this in the introduction would help the reader understand one of the critical advances of the manuscript – dynamic feedbacks – without having to go to the methods.

(L110-117) The result that common species can have weaker negative feedback than rare species because of pathogen accumulations under heterospecifics and mutualist accumulation under conspecifics is really cool! But, I'm worried that readers not immersed in the Is (Bever et al. 1997) worldview of plant-soil feedback may miss how this generates weaker negative feedback, even though it is verbally described in L121-125. Again, I hate to add length to the main body of the manuscript, but I think it might be essential to describe pairwise feedback, how it is calculated, and what it means (especially for coexistence) somewhere other than the methods.

(L312) "tree species' relative abundance in the seed bank" – I took this to mean the "seed bank" from that "year" rather than an accumulation over each time step (which is what "seed bank" evokes). Is that true or am I missing something?

(Table 1) In the table, values that are most likely to drive a positive correlation between host abundance and feedback are reported. Is there enough data available to compare the values for some of these parameters to compare them to observed values? The discussion attempts to get at this and

offer future directions for research, but it would be powerful if some of these values linked to observed values.

Minor comments:

(L63) Change "tree" to "plant" for a more general introduction

(L67) "obscure" rather than "obscuring"

(L480) Differences "in" fecundity

Reviewer #2:

Remarks to the Author:

Feedback between plants and the soil microbiome is thought to be a phenomenon that promotes coexistence of a large number of plant species within a community. But the details of the mechanisms are not understood. To advance understanding of these mechanisms, the authors model the effects of plant-soil feedbacks (PSF) as factors in plant biodiversity. They show, using a stochastic one-dimensional spatially explicit model, that both plant host abundance and the competitive abilities of soil pathogens and mutualists are important in creating the patterns observed. The model is able to explain how both the positive correlation of PSF and tree commonness and high tree diversity can occur together. Strict host specificity is not required, and dynamics of soil biota seems to be a key factor. The authors also identify, using random forest, the main parameters associated with maintenance of plant diversity, direction (positive or negative) of plant-soil feedback, and positive correlation between plant abundance and feedback.

Comments

The results here are new and an important contribution to understanding how interactions plant species and their soil microbiomes can explain plant diversity and other observed aspects of plant community and the soil biota communities. The authors make a number of assumptions; including a fixed number of individual plants, dispersal limitation assumptions, a mortality regime in which trees are removed and replaced at random, interactions between plants that are limited to plant-soil feedbacks, plant fitness differences only at the seedling level, and plants being aligned one-dimensionally. Although these assumptions are simplifying in some respects, they are not oversimplifying, and the model seems to be a reasonable start towards the questions being addressed. Clearly a great deal of thinking has gone into the design of the model.

I have a few minor questions.

Line 51. The authors note that common species generally exhibit weaker negative PSF than rare species, and their model is able to demonstrate that such a pattern could occur while at the same time high plant diversity is maintained. However, in the section in Lines 427-453 it is noted that the 'probability of randomly achieving a simulation parameterization with strong correlation between feedback strength and abundance was low.' If, as stated, the phenomenon is observed in nature, is there a reason why it is hard to find parameter values showing that in the model?

Lines 262-270. I was not quite sure of the cause of the potential slow recovery of species experiencing weak negative PSF. Is that it connected to relative high abundance of pathogens beneath heterospecifics, which can result in low seedling survival of the focal species in sites formerly occupied by those heterospecifics?

Lines 271-279. Also, the authors note that areas with selective logging over long periods develop

stronger negative feedback, whereas those in which certain plants are selectively cultivated develop weaker negative feedbacks. Is this due to gradual loss or gain of host-specific mutualists, respectively, over time, or some other cause?

The occurrence of soil pathogens and mutualists specific to a particular host that occur on the soil of heterospecifics is a key mechanism. Is there a way to determine what is the more important component of those soil biota; new input dispersing from nearby sites or the persistence of the biota after trees have died?

Do individual simulations approach some sort of stable community; i.e., roughly constant proportions of species and spatial structure? If so, does it vary much among replicate simulations?

The number of plant species is small (five). Can this easily be extended to more species; which might require extending the number of individuals beyond 499.

Are any higher order effects noticeable; e.g., the presence of one plant species having disproportionate effects on another given species?

Minor:

Line 123. Change 'mutualist' to 'mutualists'

Line 172. When mentioning h , relative effect of mutualists on seedling survival, it might be useful to refer to Equation (4), where h occurs.

Line 182. When mentioning q_f as the parameter determining how long pathogens persist at a site, and thus governs pathogen density in heterospecifics, it may be useful to mention that it occurs in equation (5).

Line 212. Change 'mutualists' to 'mutualist' or 'mutualists' '

Line 342. How many time steps are needed for a tree to become an adult?

Line 480. Change 'differences' to 'differences in'

Overall, this is important paper in advancing the understanding of how the detailed patterns observed in plants and their soil microbiome communities can be created by plant-soil feedback, using a small number of reasonable assumptions on plant and soil biota traits. This comprehensive modeling study will certainly be of great interest to a broad audience of community and ecosystem ecologists.

Thank you to both reviewers for your valuable feedback that helped us increase the quality of the manuscript.

We have italicized reviewer comments below for clarity.

Reviewer #1 (Remarks to the Author):

In this manuscript, the authors used simulation models to explore how changes in the traits of microbial pathogens and mutualists influence plant-soil feedback. Specifically, they were interested in determining what scenarios could resolve an apparent “paradox” in the world of plant-soil feedback – how to reconcile negative plant-soil feedback as a coexistence mechanism with the observation that common plant species often have weaker negative feedbacks. While most simulations of plant-soil feedback treat feedback as a fixed trait of the plant, these simulations allowed the composition of mutualist and pathogen communities to change based on composition of the surrounding microbial communities, competition among microbes, and affinity for host plants. The authors found support for negative plant-soil feedback generating coexistence among species and found that allowing different pathogen and mutualist dynamics can generate negative plant-soil feedback and the positive correlation between feedback and abundance.

Given the interest in plant-soil feedback as a coexistence mechanism and the conundrum of negative feedback generating coexistence but common species having weaker negative feedback, I suspect this paper will be of interest to many plant community ecologists. While I am not an expert on the methods, they seem sound and generated interesting, plausible results. The discussion is full of exiting ideas that can be empirically tested. My main critique is that I believe more details on plant-soil feedback and the model should be added to the introduction to let the reader follow along without having to dig into the methods.

Response: Thank you very much for taking the time to provide constructive comments and questions. Please see our responses below.

(L75) I know there are space limitations, but it would help place the work in context to provide a few key empirical examples of how mutualistic and pathogenic microbes differ in traits.

Response: We revised the text to include some examples of how mutualist and pathogen traits differ. For dispersal, direct comparisons between the groups are limited, so we provide examples of how dispersal propensity varies within each group. Thus, depending on the system (e.g. arbuscular mycorrhizal forest vs. ectomycorrhizal forest), the relative trait values of mutualists and pathogens can be expected to differ. The paragraph (L67-78) now reads:

Understanding what drives interspecific variation in feedback strength requires understanding microbial characteristics that affect population dynamics of the

mutualists and pathogens that underlie feedbacks. These traits, such as host affinity and dispersal, may differ between microbes with mutualistic and those with pathogenic effects on their hosts. For example, studies comparing host association patterns of mutualists and pathogens suggest that putative pathogens show stronger affinity for particular hosts than mutualistic arbuscular mycorrhizal fungi²²⁻²⁴. Additionally, both mutualists and pathogens can vary in dispersal propensity, from broad dispersal by wind^{25,26}, to limited dispersal via hyphal growth or belowground spores^{25,27}. Thus, depending on which types of mutualists or pathogens predominate in a given system, the two guilds can differ in relative dispersal. This potential for pathogens and mutualists to differ in life history traits underscores the importance of considering how both mutualists and pathogens independently affect feedback.

We also increased the clarity of the discussion section dealing with this topic (L224-233).

In tropical forests, where PSF has been observed to be negative⁷, plant-associated mutualists appear to be less host-specific in their association patterns than pathogens^{9,22,23}. This topic deserves more attention, ideally empirical quantification of the effects of mutualists and pathogens on host growth and survival, and how these effects differ between host plant species and environments. Theory provides some important clues as to how these traits are expected to vary between environments. For example, Thrall et al.³³ predict that pathogens more strongly affect host survival with increasing resource availability, while mutualists become increasingly host-specific (and pathogens less-so) with increasing host diversity. Unfortunately, direct comparisons between characteristics of mutualists and pathogens are rare or lacking. Everything described above emphasizes the need for these studies.

(L83) “allow mutualist and pathogens communities to dynamically change beneath each tree following competition equations” – From the introduction, it is still unclear how the simulations are allowing microbial communities to change. Earlier (L63-66) made it seem like it was going to be a function of the surrounding plant community. Here, it seems like it is a function of microbial competition. After reading the methods, it is clear that it is a function of both factors, along with the compatibility between the microbes and the host. Clearly stating this in the introduction would help the reader understand one of the critical advances of the manuscript – dynamic feedbacks – without having to go to the methods.

Response: We have clarified the language in the introduction to reflect the fact that host plant identity indirectly affects microbial community composition by affecting microbial community dynamics

We have replaced L63-66 (now L57-64) with:

It is therefore clear that the identity of the host plant *indirectly affects* the local microbial community composition by affecting the dynamics of the existing microbial community. Accordingly, factors such as the spatial arrangement of neighbouring plants, and their local species abundances, may influence the dynamics of the soil microbiota that mediate PSF^{7,13,17}. The assumption that target host plant identity *directly determines* the local soil microbiome thus obscures potentially important mechanisms underlying the relationship between PSF and plant community structure.

We also replaced L79-82 (now L81-85) with:

Departing from previous simulations, we use competition equations to control the change of the mutualist and pathogen communities through time on each host. We assume host plant identity does not *directly determine* microbial community composition; rather, plant identity affects the parameter values of the competition equations controlling microbial community dynamics.

(L110-117) The result that common species can have weaker negative feedback than rare species because of pathogen accumulations under heterospecifics and mutualist accumulation under conspecifics is really cool! But, I'm worried that readers not immersed in the Is (Bever et al. 1997) worldview of plant-soil feedback may miss how this generates weaker negative feedback, even though it is verbally described in L121-125. Again, I hate to add length to the main body of the manuscript, but I think it might be essential to describe pairwise feedback, how it is calculated, and what it means (especially for coexistence) somewhere other than the methods.

Response: Thank you for this suggestion. We now include a box to explain the feedback interaction coefficient (referenced in line 34). We also include a second box to explain how PSF strength could vary between common and rare hosts (referenced in line 49).

(L312) "tree species' relative abundance in the seed bank" – I took this to mean the "seed bank" from that "year" rather than an accumulation over each time step (which is what "seed bank" evokes). Is that true or am I missing something?

Response: Thank you for noticing this issue. We did not intend to imply that seeds accumulate through time. The sentence (L317-320) now reads:

The probability that an individual of a given tree species recruits to the adult population at a site is a function of the community of microbial propagules (pathogens and mutualists), and the relative abundance of the tree species' seeds, occurring at the site.

(Table 1) In the table, values that are most likely to drive a positive correlation between host abundance and feedback are reported. Is there enough data available to compare the values for some of these parameters to compare them to observed values? The

discussion attempts to get at this and offer future directions for research, but it would be powerful if some of these values linked to observed values.

Response: Currently, matching these results to measured values is very speculative. Nonetheless, we added some information to Table 1. Specifically, we now provide some references to empirical results that provide relevant background for several of the traits.

Minor comments:

(L63) Change “tree” to “plant” for a more general introduction

Response: We now use “plant” in the introduction until we specifically discuss the dynamics of our simulation.

(L67) “obscure” rather than “obscuring”

Response: We revised this sentence (L62-64) to read:

The assumption that target host plant identity *directly determines* the local soil microbiome thus obscures potentially important mechanisms underlying the relationship between PSF and plant community structure.

(L480) Differences “in” fecundity

Response: Corrected, thank you.

Reviewer #2 (Remarks to the Author):

Feedback between plants and the soil microbiome is thought to be a phenomenon that promotes coexistence of a large number of plant species within a community. But the details of the mechanisms are not understood. To advance understanding of these mechanisms, the authors model the effects of plant-soil feedbacks (PSF) as factors in plant biodiversity. They show, using a stochastic one-dimensional spatially explicit model, that both plant host abundance and the competitive abilities of soil pathogens and mutualists are important in creating the patterns observed. The model is able to explain how both the positive correlation of PSF and tree commonness and high tree diversity can occur together. Strict host specificity is not required, and dynamics of soil biota seems to be a key factor. The authors also identify, using random forest, the main parameters associated with maintenance of plant diversity, direction (positive or negative) of plant-soil feedback, and positive correlation between plant abundance and feedback.

Comments

The results here are new and an important contribution to understanding how interactions plant species and their soil microbiomes can explain plant diversity and other observed aspects of plant community and the soil biota communities. The authors

make a number of assumptions; including a fixed number of individual plants, dispersal limitation assumptions, a mortality regime in which trees are removed and replaced at random, interactions between plants that are limited to plant-soil feedbacks, plant fitness differences only at the seedling level, and plants being aligned one-dimensionally. Although these assumptions are simplifying in some respects, they are not oversimplifying, and the model seems to be a reasonable start towards the questions being addressed. Clearly a great deal of thinking has gone into the design of the model.

Response: Thank you very much for your review. We appreciate the constructive feedback and hope we have adequately addressed your concerns.

I have a few minor questions.

Line 51. The authors note that common species generally exhibit weaker negative PSF than rare species, and their model is able to demonstrate that such a pattern could occur while at the same time high plant diversity is maintained. However, in the section in Lines 427-453 it is noted that the ‘probability of randomly achieving a simulation parameterization with strong correlation between feedback strength and abundance was low.’ If, as stated, the phenomenon is observed in nature, is there a reason why it is hard to find parameter values showing that in the model?

Response: If we focus on the subset of simulations that produce biologically realistic scenarios (negative PSF and coexistence of all five plant, pathogen, and mutualist, species), the positive correlation between host abundance and feedback strength is a relatively more common result. In fact, it is more common than a negative correlation between PSF and host abundance. We revised the manuscript to more clearly explain why we used the optimization approach (L450-458).

The probability of randomly achieving a simulation parameterization that matched any set of PSF patterns was low, because we explored a very broad range of parameter values. This broad search allowed us to clearly understand the independent influence of each parameter on simulation results using random forest models, but also meant that any specific simulation result was poorly replicated in our random search. We therefore used particle swarm optimization (PSO) to identify combinations of parameter values most likely to simultaneously promote species coexistence under negative PSF, and generate a positive correlation between PSF and plant abundance (i.e. $\rho(I_s) > 0.8$, $\sigma(I_s) > 0.02$, $I_{max} < 0$, and richness of plants, mutualists, and pathogens = 5).

Lines 262-270. I was not quite sure of the cause of the potential slow recovery of species experiencing weak negative PSF. Is that it connected to relative high abundance of pathogens beneath heterospecifics, which can result in low seedling survival of the focal species in sites formerly occupied by those heterospecifics?

Response: Your interpretation is correct. However, we decided that the previous paragraph (now L264-272) already communicates the importance of determining whether variation in PSF strength results from variation in survival beneath conspecifics vs. heterospecifics. We therefore decided to increase the clarity of the manuscript by removing this paragraph.

Lines 271-279. Also, the authors note that areas with selective logging over long periods develop stronger negative feedback, whereas those in which certain plants are selectively cultivated develop weaker negative feedbacks. Is this due to gradual loss or gain of host-specific mutualists, respectively, over time, or some other cause?

Response: Given that these processes would likely require long periods of time to take effect, we decided to increase the clarity of the manuscript by removing this passage.

The occurrence of soil pathogens and mutualists specific to a particular host that occur on the soil of heterospecifics is a key mechanism. Is there a way to determine what is the more important component of those soil biota; new input dispersing from nearby sites or the persistence of the biota after trees have died?

Response: Yes. According to the random forest model variable importance plot (Fig S3), the persistence of pathogens on heterospecific hosts (conferred by q_p) is much more important than the abundance of pathogen spores dispersing from nearby hosts, γ_p , in determining whether a positive correlation between host abundance and PSF develops. We now explicitly make this point in the results section (L192-196):

Notably, q_p is much more important as a predictor of a positive correlation between host abundance and PSF strength than pathogen dispersal distance, b_p (Eq. 3b), or fecundity, γ_p (Eq. 3a). This shows that pathogen persistence, rather than more recent spill-over from nearby common hosts, primarily drives the correlation between host abundance and PSF strength.

Do individual simulations approach some sort of stable community; i.e., roughly constant proportions of species and spatial structure? If so, does it vary much among replicate simulations?

Response: The simulations do tend to reach stable equilibrium abundances under strong negative feedback. Although these equilibrium abundances are different between species (rank abundance of plants is determined by plant fitness), replicate simulations reach the same stable equilibria. Figure 3a illustrates the different stable equilibria reached by three species of different fitness levels. We also added an equilibrium metric (discussed in line 409-416), which uses a correlation of abundances through time for the most abundant species to determine whether a simulation has reached equilibrium. We added a random forest model to predict whether plant dynamics reach an equilibrium, and find that simulation runs tend to reach stable equilibria under negative feedback, but not under positive feedback (Figure S2).

The number of plant species is small (five). Can this easily be extended to more species; which might require extending the number of individuals beyond 499.

Response: The simulation can be scaled up, but it is limited by computer memory and speed. With each additional plant species, an additional mutualist and pathogen is required, which must be tracked on each tree. Furthermore, to prevent stochastic extinctions of relatively more rare species, the size of the simulation also needs to be increased as species are added. Extending the simulation would therefore require a redesign so that it could run on machines with greater capacity for parallelization. Expanding the size of the simulation is beyond the scope of this manuscript, but would be very valuable to explore in the future.

Are any higher order effects noticeable; e.g., the presence of one plant species having disproportionate effects on another given species?

Response: Yes, we can see evidence of higher order effects in Fig. 3 (previously Fig. 2). We now explicitly mention this result in the manuscript (L133-137):

The dynamic mutualist and pathogen communities can also generate higher-order patterns of plant-soil feedback. Specifically, each plant species can have a different effect on the relative survival probability of the other species. For example, seedlings of the rare species are relatively more likely to survive beneath individuals of the intermediately abundant species than beneath individuals of a common species (Fig. 2b).

Minor:

Line 123. Change 'mutualist' to 'mutualists'

Response: This has been corrected, thank you.

Line 172. When mentioning h , relative effect of mutualists on seedling survival, it might be useful to refer to Equation (4), where h occurs.

Response: We now include the suggested reference (now Equation 5).

Line 182. When mentioning q_f as the parameter determining how long pathogens persist at a site, and thus governs pathogen density in heterospecifics, it may be useful to mention that it occurs in equation (5).

Response: We now include the suggested reference (now Equation 6).

Line 212. Change 'mutualists' to 'mutualist' or 'mutualists' '

Response: We changed 'mutualists' to 'mutualist'

Line 342. How many time steps are needed for a tree to become an adult?

Response: We assume that a tree becomes an adult in the next time step after it successfully recruits. We now explicitly state this after describing the recruitment process (L388-389):

The recruit becomes a reproductive adult in the next time step, and begins modifying the abundance of each microbial taxon through time according to the following differential equation:

Line 480. Change 'differences' to 'differences in'

Response: This has been corrected, thank you.

Overall, this is important paper in advancing the understanding of how the detailed patterns observed in plants and their soil microbiome communities can be created by plant-soil feedback, using a small number of reasonable assumptions on plant and soil biota traits. This comprehensive modeling study will certainly be of great interest to a broad audience of community and ecosystem ecologists.

Response: Thank you again for your time, and for your constructive feedback.

In addition to the changes outlined above, we also improved the following aspects of the manuscript:

1. Rather than measure PSF at the end of the simulation, and use the value of PSF to predict plant diversity (e.g. in the previous version of Fig. 4), we now measure feedback at the start of the simulation. Specifically, we first let the microbial communities develop for 50 time steps without any host mortality, then calculate PSF (described in L402-405). We use this value of PSF to then predict the dynamics of the simulation.
2. In Figure 1, we now display the mean values (+/- s.e.) of abundance and feedback for each species across multiple runs, rather than display species-specific PSF values for each run.
3. We removed log ratios of trait values as predictors of model dynamics, because variable interactions are already automatically incorporated in to the random forest model framework. This simplifies the interpretation of model results.

Reviewers' Comments:

Reviewer #1:

Remarks to the Author:

I thank the authors for producing thoughtful responses and edits to their manuscript. My comments have been fully addressed. The two added boxes are especially helpful additions.

Reviewer #2:

Remarks to the Author:

The revised version clarifies a number of questions I had with the original version. I have a few further minor comments.

Lines 27-28. I suggest revising 'that, in turn, differentially affect' to 'which, in turn, differentially affects' because 'affects' refers back to 'the composition of the local soil microbiome'.

Lines 118-122. This doesn't appear to be a sentence. Maybe remove the 'that'.

Line 182. 'Figureb'. I think this must be Figure 5b, as the figure shows negative feedback increasing as the host specificity increases, in agreement with the text.

Line 257. The sentence (Lines 255-257) would be a bit clearer if 'their' in Line 257 was identified as the common plant, as, at least grammatically, the 'their' could apply to the rare plant, which is the closer antecedent.

Lines 372, 385, 464. Change 'Where' to 'where'

The results of the manuscript are interesting and non-intuitive and are an important step in understanding effects of plant-soil microbiome interactions on plant community structure.

We appreciate constructive final comments from both reviewers. We have italicized reviewer comments below for clarity.

REVIEWERS' COMMENTS:

Reviewer #1 (Remarks to the Author):

I thank the authors for producing thoughtful responses and edits to their manuscript. My comments have been fully addressed. The two added boxes are especially helpful additions.

Reviewer #2 (Remarks to the Author):

The revised version clarifies a number of questions I had with the original version. I have a few further minor comments.

Lines 27-28. I suggest revising 'that, in turn, differentially affect' to 'which, in turn, differentially affects' because 'affects' refers back to 'the composition of the local soil microbiome'.

Response: This has been revised with track changes.

Lines 118-122. This doesn't appear to be a sentence. Maybe remove the 'that'.

Response: We agree. Removing 'that' resolves the issue.

Line 182. 'Figureb'. I think this must be Figure 5b, as the figure shows negative feedback increasing as the host specificity increases, in agreement with the text.

Response: This, and similar errors, were introduced when the document was converted from an MS word file to a PDF. We have fixed this issue.

Line 257. The sentence (Lines 255-257) would be a bit clearer if 'their' in Line 257 was identified as the common plant, as, at least grammatically, the 'their' could apply to the rare plant, which is the closer antecedent.

Response: We agree. We have modified the sentence to read:

As plants become more common, their pathogens become relatively more abundant beneath heterospecifics than those of rare plants, thereby reducing the strength of net negative feedback for common plants, and reducing their survival probability.

Lines 372, 385, 464. Change 'Where' to 'where'

Response: Done

The results of the manuscript are interesting and non-intuitive and are an important step in understanding effects of plant-soil microbiome interactions on plant community structure.